# Theta Band (4–8 Hz) Oscillations Reflect Online Processing of Rhythm in Speech Production

**DOI:** 10.3390/brainsci12121593

**Published:** 2022-11-22

**Authors:** Qin Yan, Qingfang Zhang

**Affiliations:** 1Department of Psychology, Renmin University of China, Beijing 100872, China; 2Laboratory of Department of Psychology, Renmin University of China, Beijing 100872, China; 3Interdisciplinary Platform of Philosophy and Cognitive Science, Renmin University of China, Beijing 100872, China

**Keywords:** speech rhythm, language production, phrase-recall paradigm, cerebral-acoustic coherence (CACoh), theta band oscillation

## Abstract

How speech prosody is processed in the brain during language production remains an unsolved issue. The present work used the phrase-recall paradigm to analyze brain oscillation underpinning rhythmic processing in speech production. Participants were told to recall target speeches aloud consisting of verb–noun pairings with a common (e.g., [2+2], the numbers in brackets represent the number of syllables) or uncommon (e.g., [1+3]) rhythmic pattern. Target speeches were preceded by rhythmic musical patterns, either congruent or incongruent, created by using pure tones at various temporal intervals. Electroencephalogram signals were recorded throughout the experiment. Behavioral results in 2+2 target speeches showed a rhythmic priming effect when comparing congruent and incongruent conditions. Cerebral-acoustic coherence analysis showed that neural activities synchronized with the rhythmic patterns of primes. Furthermore, target phrases that had congruent rhythmic patterns with a prime rhythm were associated with increased theta-band (4–8 Hz) activity in the time window of 400–800 ms in both the 2+2 and 1+3 target conditions. These findings suggest that rhythmic patterns can be processed online. Neural activities synchronize with the rhythmic input and speakers create an abstract rhythmic pattern before and during articulation in speech production.

## 1. Introduction

Speech has distinct rhythmic properties, and this rhythmicity may be central to how we process language. Speech rhythm is the temporal structure of auditory events that occur across time [1,2]. As rhythm is consistently aligned with the speech structure, it is useful for segmenting speech [3,4,5], enhancing linguistic processing when speech rhythm is predictable [1,6], and facilitating speech production in some cases [7,8]. Over the last decade, researchers have explored the cognitive brain mechanisms that underlie speech rhythmic structure. As a result, the time course and the brain regions underlying the speech rhythmic processing have been elucidated, particularly in the area of language perception and comprehension (e.g., [1,9,10,11], see [5] for a review). However, so far, no studies have examined the brain oscillations of rhythmic information in language production and the current work is carried out to fill this gap.

How people actually construct language in real time is the central query that language production theories aim to answer. Speaking is made up of three major types of processes: Conceptualization, formulation, and articulation [12]. Conceptualization processes generate messages while formulation processes encode grammatical and phonological information. Then, the above information is used to retrieve and run articulatory programs. Researchers point out that these stages are processed incrementally and in parallel, which implies that stages need not complete their work on an utterance before the next stage begins [12]. As for prosody, a number of studies have reported that prosody influences the planning phase of spoken language production [8,13,14]. Nevertheless, some researchers argue that speech prosody is not produced at a certain stage in speech production and the processing of prosody may go through all stages of speech production until the utterance is finished [15]. In this study, we concentrate primarily on the rhythmic aspect of prosody. We attempt to explore the stages of rhythmic processing in Chinese speech production and neural oscillations related to the rhythmic priming effect by manipulating congruous and incongruous rhythmic conditions between primes and targets. More specifically, we want to know if a simple pure tone pattern could be used to prime rhythmic information in Chinese speech production using the phrase-recall paradigm. This study has two goals: To find out if brain oscillations are in sync with external rhythmic stimuli and to find out if the rhythmic priming effect affects speech-related theta-band activities.

## 2. Literature Review

### 2.1. Structural Priming for Prosodic Representations

In recent decades, researchers have utilized structural priming to identify some of the representations humans generate when producing or interpreting language [16]. Researchers contend that priming occurs at all levels of processing during language comprehension and production and that the lack of priming implies the absence of a matching level of representation [16]. Priming has been found not only for simple representations, such as lexical priming [17] but also for more complicated representations, such as the building of syntactic structures [18] and the building of a situation model [19]. 

Priming tasks have also been used to determine if abstract prosodic representations exist in language production using behavioral and electrophysiological measures. Results show that some features of prosody, such as speaking tempo and speech rhythm [8,13,20], appear to be more receptive to priming than others, such as phrase boundaries and pitch accenting [13,14]. Behavioral results show that speaking rate (e.g., [13,20]) may carry over from one sentence to the next. Jungers and Hupp [20] investigated the durability of the rate in unscripted picture description production. In their experiment, participants heard and repeated priming sentences that were spoken quickly or slowly. Then participants were asked to describe a new picture. The results showed that when speakers were given a fast sentence before describing a picture, the descriptions were faster than when they were given a slow sentence. This finding was replicated by Tooley et al. [13] in a reading-aloud task. 

Behavioral results show that the priming effect also exists in speech rhythm in language production [8,21,22]. For example, Gould et al. [22] investigated whether a rhythmic prime that was either congruent or incongruent with the target syllabic stress influenced reading aloud. They found faster reaction times for target syllables that had the same rhythmic pattern as the preceding rhythmic prime. Similarly, Zhang and Zhang [8] also employed the reading-aloud task to determine if a specific rhythmic pattern could influence the planning stage of speech production by using Chinese materials. In accordance with Gould et al.’s [22] findings, they found that reading latencies were faster when the prime and target rhythmic patterns were congruent. The rhythmic priming effects are also supported by data from the hearing-impaired population [21]. Overall, all these findings indicate that abstract representations of speech rate and speech rhythm may be formed in language production before articulation.

Nevertheless, researchers found that several features of prosody are not primeable, including phrase boundaries and pitch accenting [13,14]. Tooley et al. [14] altered the presence of a boundary at two syntactic positions in prime sentences that were auditorily delivered to participants. The speakers were required to repeat the prime sentences and silently read and verbally recite a target sentence that was visually displayed to them. Their results showed that the priming effect did not carry over to the target sentence. In their following studies [13], this finding was replicated. Additionally, Tooley et al. [13] used a similar paradigm to examine speakers’ sensitivity to pitch accenting. Once more, they found no pitch priming. Together with previous studies, we can see that different features of prosody share different types of underpinning representation as they are not equally primeable. Priming may be more easily observed in a paralinguistic element of prosody (such as the speaking rate and rhythm) than in linguistic features of prosody (such as intonational phrase boundaries and pitch accenting).

### 2.2. Electrophysiological Studies of Rhythmic Processing in Language 

The electrophysiological (EEG) technique is well known for providing high-temporal-resolution measures. Important research has focused on the temporal course of rhythmic processing in language comprehension, perception, and production by using the event-related potentials (ERPs) technique (e.g., [1,8,10,23]. Numerous research studies in the area of language comprehension have shown how rhythmic cues are immediately used in sentence processing by identifying particular ERP components. When violating the rhythmic rules, it may elicit a positive-going or/and negative-going waveform in ERP responses (e.g., P200: [24,25]; LAN: [26,27,28]; N400: [10,23]; P600: [27,28]).

The ERP approach has also been applied to explore rhythmic processing in speech perception and production. Cason and Schön [1] investigated whether the prior presentation of a rhythmical prime would improve speech processing. The results showed that in the time window of 100–500 ms, metrical mismatches evoked a greater N100 amplitude than metrical matches did, possibly indicating a violation of rhythmical expectations. Moreover, they also found that in the time range of 250–500 ms (P300), the target off-beat condition elicited a larger positive ERP component than on-beat conditions, which was associated with beat expectancy in phonological processing. On the basis of Cason and Schön work [1], Zhang and Zhang [8] also found a rhythmic priming effect in language production by using Chinese materials in a reading-aloud task. An early component (N100) was triggered when the rhythm of the target speech did not match the rhythm of the primes. Researchers argued that this component was associated with the recognition of hierarchical linguistic components. In addition, a later negative component (N400) was also detected, which was thought to show that the rhythmic pattern of speech production did not match what was expected.

Even though the rhythmic effect on language processing has already been noted in previous studies, researchers are just starting to determine how it works. The dynamic attending theory gives a good explanation for how speech rhythm affects speech processing [29,30,31]. This theory states that when people pay attention to an external rhythm, the phase and period of brain oscillations become coupled to the external rhythm. The coupling process is also referred to as entrainment [32]. Recent research has supported the dynamic attending theory by demonstrating that brain oscillatory responses to the rhythmic structure exist in both the music and speech domains (e.g., [33,34,35]), and the oscillations continue at the stimulus frequency long after the stimulus has ended [36]. Based on the dynamic attending theory, in our current study, we assume that brain activities synchronize with the rhythmic patterns of primes (measured by cerebro-acoustic phase coherence, CACoh, see the EEG analysis section), which induces temporal expectations via oscillations, and as a consequence, target generation in speech production is facilitated.

To summarize, a series of behavioral and ERP experiments have demonstrated that rhythmic information could influence language comprehension and production. However, behavioral studies are limited in what they can tell us about the processes underlying the generation of prosodic structure; though traditional ERP analysis is able to resolve the inner mechanism of rhythm by providing a high temporal resolution, it still cannot illustrate the neural oscillations in language comprehension and production. The time–frequency (TF) analysis can directly reflect the activation/inhibition of neurons at various activity rates and can disclose the non-phase-locked neural activity that is masked in the traditional ERP analysis [37]. The goal of this work is to explore neural oscillations of rhythm during language production in native Chinese speakers by means of electroencephalography (EEG) time–frequency analysis.

### 2.3. The Relationship between Neural Oscillations and Rhythmic Processing

Brain activities synchronize with the rhythm of a variety of environmental inputs, and the ability of neural activities to entrain the rhythm of speech is crucial for effective communication [34,38]. Therefore, understanding brain oscillations may offer a plausible mechanistic explanation for how speech rhythm is processed in the brain [39,40].

Neural oscillations can reflect the rhythmic activity of neurons in the brain, including delta (δ, <4 Hz), theta (θ, 4–8 Hz), alpha (α, 8–13 Hz), beta (β, 13–30 Hz), and gamma (γ, >30 Hz) [34,41]. Studies have shown that neural oscillations are crucial for cognitive functions such as memory, attention, and temporal prediction [42,43,44,45]. Neural oscillations are also essential in language processing ([40,46], see [47] for a review). The entrainment of brain oscillations to speech amplitude rhythms has been observed at the gamma [48,49], theta [50,51], and delta frequencies [48,52]. These electrophysiological rhythms are assumed to be entrained because their frequencies match those of the rhythmic amplitude edges and peaks that come with phonemes, syllables, and intonation phrases. According to oscillation-based theories, speech rhythm in language processing appears to be primarily dependent on low-frequency brain oscillations, notably for theta-frequency bands ([34], see [53] for a review).

Emerging evidence from neurophysiological studies indicates that the theta-band (4–8 Hz) activity phase locks to the speech rhythm in speech perception (e.g., [40,54,55]). For instance, Park et al. [56] discovered that theta-band brain oscillatory activity is associated with the perception of rhythmic boundaries. They observed the brain oscillations of 22 participants during continuous speech perception using magnetoencephalography (MEG). Participants were required to listen to the story speech for 7 min. Then they computed transfer entropy (TE) time-resolved and centered on rhythmic boundaries (‘‘edges’’) in the continuous speech. Results show that theta TE increased in the left inferior frontal and precentral gyri before and around the commencement of rhythmic boundaries. In line with this finding, Power et al. [57] used a rhythmic entrainment paradigm and also found evidence of theta-band neural oscillation for speech rhythm. They used repetition of the syllable “ba” and asked participants to respond by pressing a button as quickly as they could when the “ba” broke the rhythm. The rhythmic violation was manipulated by postponing the appearance of “ba” in the isochronous stream. Results showed that theta-phase consistency increased when participants were exposed to an isochronous stream of syllable sequences. All these studies suggest that theta-band activities are vital in the perception of speech rhythm. Nevertheless, unlike speech perception, which has been heavily investigated, no research has been dedicated to speech rhythm in language production.

Overall, both behavioral and ERP studies showed that speech rhythm can be represented in speech production, and evidence from language perception indicated that brain activities in the theta band are closely associated with speech rhythmic processing. At present, there is still a lack of research investigating whether the rhythmic priming effect in Chinese spoken production is related to specific neural oscillations, and understanding the spectral characteristics of neural oscillation can shed light on the mechanisms of rhythmic processing in language production.

### 2.4. The Present Study

Prior research studies have typically used stress-timed languages (such as English or German) to investigate the impact of rhythm on speech processing. In stress-timed languages, stress is perceived as the alternation of strong and weak accents, which are realized by differences in amplitude, duration, and fundamental frequency [58]. The rhythmic pattern in Chinese is the combination of words with various syllable lengths [23]. The vast majority of words in contemporary Chinese are either monomorphemic, monosyllabic, or disyllabic [59]. Chinese is a syllable-timed language, thus each syllable in speech takes approximately the same amount of time. Linguistically, native Chinese speakers prefer to combine two syllables in utterances, a process known as disyllabification [60]. Under this preference, speakers prefer to split a four-word phrase or sentence into two disyllabic words (for detailed information, see the Materials section). Therefore, within a four-word phrase or sentence, a [2+2] rhythmic pattern (numbers in brackets indicate the number of syllables) is common, whereas a [1+3] rhythmic pattern is uncommon [60]. 

This research aims to further understand the brain activity involved in rhythmic processing in Chinese four-word phrase production by using a phrase-recall paradigm to investigate whether the rhythmic priming effect still exists during Chinese language production, which was observed in our previous research [8]. In our earlier research, the rhythmic priming effect was detected via a reading-aloud task. In that experiment, firstly, a musical-like prime was auditorily represented and then participants were instructed to verbalize the four-word target sentences/phrases when they saw the target speeches. As target speeches were visually presented for participants, we could not separate the comprehensive process from language production in that task. Therefore, we adopted a modified version of the phrase-recall priming paradigm in this study, which could lessen the involvement of the comprehensive process to a large degree.

In the current experiment, a target four-word phrase, which participants needed to repeat back in the later production stage, was presented to participants while EEG signals were recorded concurrently. Then we used music-like pure tones as primes, which were auditorily presented to participants [1,8,61]. The target speech’s rhythmic pattern was either consistent or incongruent with the prime’s rhythmic pattern. We postulated that a rhythmic pattern resembling music might increase listeners’ anticipation of the target utterances. Finally, participants were instructed to verbally recall the target phrase from memory. During the recall stage, we explored whether the rhythmic structure from the priming event persisted during the recall phase.

For the statistics, we first set out to investigate the reaction latency of the target rhythm and determined whether there were distinct changes in reaction latency for processing incongruent rhythmic patterns. We anticipate observing a facilitation effect when comparing the congruent prime and target rhythmic patterns to their incongruent counterparts. Then we identified the oscillatory brain activity when hearing particular types of priming rhythm and the oscillatory activities reflecting the processing of the target rhythmic pattern. For oscillatory neural activities of the priming rhythm, we expect that brain oscillations are synchronized to external rhythmic stimuli, which have a highly predictable rhythmic pattern [33,34]. For oscillatory neural activities of target rhythm, because the theta-band neural oscillation is strongly associated with rhythmic processing [53,56,57], we expect a significant rhythmic priming effect, and accordingly, theta-band power will show a significant difference between congruous and incongruous rhythmic conditions between primes and targets.

## 3. Materials and Methods

### 3.1. Participants

G*power analysis and our prior study [8] determined the number of participants in this study [8]. To ensure 80% power to find a Cohen’s effect of f = 0.40, the sample size had to be no less than 23. Based on the G*power analysis program of a 3 × 2 repeated-measure ANOVA [62], this value is a large effect size. A total of twenty-nine native speakers of Mandarin Chinese from the Renmin University of China (11 men and an average age of 22) participated in the study. They were right-handed people with self-reported normal hearing and normal or corrected-to-normal eyesight. They reported no medical, neurological, or mental health history. None of them had any formal musical training. Participants were given informed consent and were paid for participation.

### 3.2. Materials

We employed a series of pure tones with various temporal intervals as primes for our prime materials. In Adobe Audition CS6, four identical pure tones with a frequency of 900 Hz were generated. Each primary audio sample consisted of a rhythmic pattern of four tones that was repeated five times. In our former research, the prime was repeated three times. We increased the repetition times because it might be simpler for the uncommon (1+3) rhythmic patterns to build brain or motor synchronization. Three types of rhythmic patterns were employed as primes: “2+2”, “1+3”, and random rhythmic patterns (see Figure 1a). Researchers found that syllable duration is well established across languages for approximately 200 ms [5]. Therefore, in our experiment, each tone lasted 250 ms. For the “2+2” prime type, the distance between the first tone’s offsets and the second tone’s onsets and between the third tone’s offsets and the fourth tone’s onsets was 150 ms, but the interval between the second tone’s offsets and the third tone’s onsets was 300 ms (see Figure 1a). This pattern is referred to as “2+2”, where “2” represents two identical pure tones separated by a short period (150 ms) and “+” represents a lengthy interval (300 ms) between two tones. For the “1+3” prime type, the interval between the first tone’s offsets and the second tone’s onsets was 300 ms, whereas the gap between the second tone’s offsets and the third tone’s onsets and the fourth tone’s offsets was 150 ms, resulting in a “1+3” rhythmic pattern (see Figure 1a). Each of these two rhythmic patterns was repeated five times for a total duration of 1600 ms. The time between each iteration of the pattern was 600 ms, making the total primes time 10.4 s. For the random rhythmic type, the interval between the former tone’s offset and the latter tone’s onset was random and lasted a total of 10.4 s. 

For target speeches, we used verb–noun phrases consisting of four Chinese characters as the materials that needed to be recalled aloud in the experiment. The rhythmic patterns for speeches were [2+2] or [1+3] patterns (numbers in brackets denote the number of syllables). For example, in the phrase 打扫房间 (“clean the room”), the speakers typically divided the first two characters and the last two characters into two different words: 打扫 (“clean”) and 房间 (“the room”). This speech pattern was described as [2+2], with a lengthy pause between the second and third words but a relatively short pause between syllables inside a word. The same logic also applies to the [1+3] speech pattern. For instance, the phrase 修自行车 (“fix a bike”) was typically broken up into two words: 修 (“fix”) and 自行车 (“a bike”). This speech pattern was described as [1+3], with a long interval between the first two words and a comparatively short pause between the final three syllables inside a word. There were 390 target speeches in total, with 195 for the [2+2] speech pattern and 195 for the [1+3] speech pattern. 

Prior to the formal experiment, 20 people who were not in the EEG study rated how familiar target phrases were to them. A seven-point scale was used to rate participants’ familiarity with the target phrases, with “1” denoting a high level of unfamiliarity and “7” denoting a high level of familiarity. Results revealed no statistically significant differences in phrases familiarity (*t* (388) = 1.162, *p* = 0.246), indicating that the [2+2] (*Mean* = 5.933, SD = 0.268) and [1+3] speech conditions (*Mean* = 5.901, SD = 0.289) were well matched in familiarity. Moreover, these 20 subjects were also required to segment the rhythmic patterns of target phrases. Results showed that there was high agreement for the rhythmic patterns of [2+2] and [1+3] target phrases, with 94.77% agreement that [2+2] target phrases had a greater interval between second-syllable offsets and third-syllable onsets and 96.95% agreement that [1+3] target phrases had a greater interval between first-syllable offsets and second-syllable onsets. The high agreement for the rhythmic patterns of target phrases indicated that the manipulation of the rhythmic patterns of target phrases was successful. Furthermore, there was no significant difference in stroke numbers between the two speech patterns (*t* (388) = 0.864, *p* = 0.388), indicating that the [2+2] (*Mean* = 32.631, SD = 6.249) and [1+3] speech patterns (*Mean* = 32.082, SD = 6.295) were well matched in visual complexity.

### 3.3. Design

The experiment manipulated factors of prime rhythmic patterns (2+2, 1+3, and random) and target rhythmic patterns (2+2 and 1+3): A 3 × 2 within-participants factorial design. A total of 195 target utterances with a 2+2 rhythmic pattern were coupled with three types of primes, yielding 65 trials for each condition. The same pairing approach was used on the remaining 195 target speeches with a 1+3 rhythmic pattern. The experiment also contained 65 fillers, which were words with two or three characters. Therefore, each participant completed a total of 455 trials, including 6 different conditions. Each participant received a total of 35 trials—including 6 different conditions—during each experimental block. A block also contained five two- or three-character fillers. The order of trials inside a block was pseudo-randomized to ensure that identical conditions were separated by at least three trials. For each participant, a different sequence was created. There were 13 blocks in total.

### 3.4. Apparatus

E-Prime Professional Software was used to conduct the experiment (Version 3.0; Psychology Software Tools, Pittsburgh, USA). Participants sat in a quiet room that was shielded from sound and electricity. Prime audios were delivered by PHILIPS headphones (SHM7410). The target presentation’s naming latencies were recorded using a voice key connected to the computer through a PST Serial Response Box. Using Neuroscan 4.3 software, EEG data were captured using 64 electrodes placed at the typical 10–20 scalp sites and protected by an elastic cap (Electro Cap International, Charlotte, USA).

### 3.5. Procedure

The current study used a modified version of the phrase-recall paradigm (Meijer & Tree, 2003) to explore rhythmic processing in language production. Participants were tested individually. After five practice trials, the formal experiment began. Each experimental trial comprised the following steps (see Figure 1b). A 600 ms fixation (+) at the center of the screen first appeared, then the screen went blank for 500 ms. The four-word goal phrase then appeared, giving participants 1000 ms to read it. After the four-word goal phrase disappeared, a 600 ms fixation (+) appeared, then a blank screen for 500 ms. Then, in the middle of the screen, an image of a speaker emerged, and the audio from the prime was played through headphones at the same time. Participants were instructed to look at the speaker’s image while they were listening to the voice. Following that, a 600 ms fixation (+) was shown on the screen, and then the screen was blank for 400–1000 ms. A question mark then appeared, and participants were instructed to recall aloud target phrases they had seen as accurately and as quickly as possible, and the question mark disappeared as soon as they responded. After they finished recalling the phrases or if there was no response to the target beyond 3000 ms, it would go to a blank screen. In this blank screen, experimenters pressed certain keys on the keyboard to judge the correctness of their answers. The following trial would start as soon as the experimenter pressed the key. Participants were permitted to take short breaks whenever necessary. The whole EEG experiment took approximately 4 h to complete.

### 3.6. EEG Recordings

With the addition of two mastoid electrodes, 62 scalp electrodes set on an elastic cap according to the extended 10–20 system were used to constantly record the electroencephalogram. The EEG data were amplified (bandpass 0.01–70 Hz) and sampled at 500 Hz. Bipolar horizontal and vertical electrooculograms (EOG) were obtained from electrodes positioned beneath the canthus of both eyes as well as above and below the right eye to monitor eye movements. The reference during the online recording was the left mastoid. Offline, the average of the left and right mastoids was used to re-reference the EEG data. Throughout the experiment, the impedance between the electrode and the scalp was kept below 5 kΩ.

### 3.7. EEG Analysis

#### 3.7.1. CACoh Preprocessing and Calculation

To determine the cortical tracking of the rhythm of priming stimuli, the analysis was time-locked to the acoustic onset of the priming stimuli. As previous studies have shown that brain activity is phase-locked to the rhythms of speech and music [33,34], we used the cerebral-acoustic coherence (CACoh; [50,63]) with the priming stimulus envelopes to determine whether the neural activity synchronizes with rhythmic patterns of primes. As we mainly focused on the common (i.e., 2+2) and uncommon (i.e., 1+3) rhythmic patterns, we measured the CACoh values of the two rhythmic patterns. 

The EEG and CACoh data were assessed using EEGLAB [64] and the FieldTrip toolbox for MATLAB 2017b [65]. A Hamming windowed sinc FIR (finite impulse response) filter was used to band-pass filter the raw EEG data (0.5–30 Hz). Offline, the EEG data were down-sampled to 100 Hz. The length of each trial corresponded to the length of the priming stimulus (10.4 s), and the continuous EEG data were epoched from 0.2 s preceding the onset of the priming stimuli to 10.4 s after their onset. The fastica independent components analysis (ICA) toolbox and the artifacts detecting the plug-in ICA-Label in MATLAB were used to exclude eye blinks, horizontal eye movements, noisy electrodes, and other physiological artifacts such as heartbeats. Noisy epochs were rejected using a ±100 μV threshold, and trials with incorrect responses, naming latencies that were faster than 300 ms or slower than 1500 ms, or those that deviated more than 2 SD from the cell mean were eliminated from further EEG analysis. Overall, 83% of trials remained, and these data were baselined using a pre-stimulus baseline. Trials were epoched into 10-s segments (+0.4 s to +10.4 s with respect to item onset). Note that the epoch began at 0.4 s in order to prevent the N1-P2 complex from contaminating onset neuronal responses [33,66]; 10.4 s corresponded to the priming stimulus length.

For cerebro-acoustic phase coherence calculations, the Hilbert transform was utilized to extract the amplitude envelopes of the acoustic priming stimuli, which were then low-pass filtered at 30 Hz and high-pass filtered at 0.5 Hz. Stimuli were zero-padded to match EEG epochs of 10.4 s. The EEG and stimulus epochs were then transformed to the time–frequency domain using wavelets linearly spaced from 3 to 7 cycles over 0.5–30 Hz with a frequency resolution of 0.5 Hz and a temporal resolution of 10 ms. Neural data were evaluated across all frequencies and electrodes [34,67]. Using the complex values derived from EEG data and stimuli, cerebro-acoustic phase coherence was estimated (see Harding et al. [33] for the formula). This phase coherence measure was calculated by comparing the phase alignment of the pre-processed EEG signal to the amplitude envelope of the corresponding sound stimuli. Cerebro-acoustic coherence was averaged over time lengths corresponding to the range of the stimulus. Cerebro-acoustic phase coherence ranges from 0 (no coherence) to 1 (complete coherence).

#### 3.7.2. TFR (Time–Frequency-Representations) Preprocessing and Calculation

To further examine the brain oscillations of the rhythmic processing during target phrases production, the EEG data were preprocessed and time–frequency representations (TFRs) of neural oscillatory power were calculated at the onset of the question mark where participants were instructed to recall aloud the target phrase, using EEGLAB [63] and the FieldTrip toolbox for MATLAB 2017b [65]. A Hamming windowed sinc FIR (finite impulse response) filter was used to band-pass filter the raw EEG data (0.5–30 Hz) and divide the epochs ranging from 1 s preceding the onset of the question mark to 1 s after their onset. The fastica independent components analysis (ICA) toolbox and the artifacts-detecting plug-in ICA-Label in MATLAB were used to exclude eye blinks, horizontal eye movements, noisy electrodes, and other physiological artifacts such as heartbeats. Noisy epochs were rejected using a ±100 μV threshold, and trials with incorrect responses, naming latencies that were faster than 300 ms or slower than 1500 ms, or those that deviated more than 2 SD from the cell mean were eliminated from further EEG analysis. Overall, 80% of trials remained and were used for further time–frequency analysis. Fast Fourier transforms (FFTs) were applied to each 2 s clean epoch in a step of 10 ms. A set of 28 log-spaced frequencies spanning from 0.5 Hz to 30 Hz in 1–Hz steps was chosen. The analysis first estimated the power activity of a single trial, and then the average of multiple trials was calculated. Power values were normalized with respect to an 800–200 ms pre-exclamation onset baseline and transformed into a decibel scale (dB = 10 × log10 activity/baseline mean activity). 

For statistical analysis, in accordance with prior research [57,67] and the topographies of our data (Figure 2), we evaluated theta power at a frontal electrode pool for statistical analysis. Theta oscillation power in this region was averaged. We adopted a 3 (rhythmic patterns of prime type) × 2 (rhythmic patterns of target type) repeated-measure ANOVA. The *p* values were corrected by the Greenhouse–Geisser method when the spherical assumption was not satisfied.

## 4. Results

### 4.1. Behavioral Results

Trials with incorrect answers and voice key errors (less than 1%) and naming latencies faster than 300 ms or slower than 1500 ms (6.86%) or trials more than 2 SD from the mean (4.82%) were eliminated. 

We employed a linear mixed-effect model (LMM) to examine the rhythmic priming effect using the lme4 package [68] in R. The outcome variable was trial-by-trial RTs for each participant. The prime rhythm, target rhythm, and their interaction were introduced as the fixed effect in the LMM, and the random interception of each participant and each item were introduced as the random effect in the LMM. LMM analysis found that there was a main effect of prime rhythm, but there was no effect of target rhythm, nor was there an interaction (see Table 1). 

We performed multiple comparisons between the target conditions for each prime rhythm pattern by using Bonferroni corrections since researchers have suggested that it is appropriate to make individual comparisons without relying on a significant omnibus F test [69]. Results revealed that, for the primes with the 2+2 rhythmic pattern, the difference between 2+2 and 1+3 targets was significant (*β* = −14.56, *z* = −2.318, *p* = 0.021), with 2+2 target latencies significantly shorter (*Mean* = 560 ms, SD = 118) than 1+3 target conditions (*Mean* = 575 ms, SD = 114) (see Figure 2). For the primes with the 1+3 and random rhythmic patterns, there were no significant differences between 2+2 and 1+3 target latencies (*p*s > 0.330).

### 4.2. CACoh and TFR Results

Results of CACoh showed that cortical tracking was not significantly different between the “1+3” and “2+2” rhythm of priming stimuli (1+3 rhythm prime: *Mean* = 0.467, SD= 0.012; 2+2 rhythm prime: *Mean* = 0.465, SD = 0.011; *t* (28) = 0.820, *p* = 0.419), indicating a similar medium-phase-locked cortical response to the “1+3” and “2+2” rhythm of priming stimuli.

For time–frequency analysis, frontal theta power was calculated by means of a 3 × 2 ANOVA with the factors “Prime rhythm” and “Target rhythm”. Figure 3 shows the average TFRs of theta power for different conditions. This analysis yielded a main effect of “Prime rhythm” (F (2,56) = 4.766, *p* = 0.012) as well as a significant “Prime rhythm” × “Target rhythm” interaction (F (2,56) = 6.286, *p* = 0.003). The post-hoc pairwise comparisons revealed that, for primes with the 1+3 rhythmic pattern, the difference between 2+2 and 1+3 targets was significant (F (1,28) = 4.689, *p* = 0.039), with theta power higher in the 1+3 target rhythm condition (*Mean* = 0.987, SD = 1.213) than that of the 2+2 target rhythm condition (*Mean* = 0.773, SD = 0.848). For primes with the 2+2 rhythmic pattern, the difference between 2+2 and 1+3 targets was significant (F (1,28) = 9.510, *p* = 0.005), with theta power higher in the 2+2 target rhythm condition (*Mean* = 0.993 dB, SD = 1.037) than that of the 1+3 target rhythm condition (*Mean* = 0.801 dB, SD = 0.895). For the primes with random rhythmic patterns, there were no significant differences between 2+2 and 1+3 targets (F (1,28) = 0.186, *p* = 0.669) (see Figure 3g).

## 5. Discussion

This research aimed to investigate the neural oscillation involved in speech rhythmic processing during language production by using the phrase-recall paradigm. In our study, a pure-tone rhythmic prime was employed to elicit expectations regarding the target speech’s rhythmic pattern. After repeated exposure to primes, participants were instructed to recall the target phrases aloud from memory. We replicated the rhythmic effect that has been found in the common rhythmic condition (i.e., 2+2 rhythmic pattern) in behavioral and electrophysiological data [8]; by using CACoh, we observed that brain activity synchronizes with the rhythmic patterns of primes, and the cortical response to pure tones was comparable in terms of common and uncommon priming rhythmic patterns (i.e., 1+3 rhythmic pattern), which was in accordance with the dynamic attending theory; furthermore, we found that target phrases with congruent rhythmic patterns were associated with the theta (4–8 Hz) increase in the 400–800 ms time window over frontal electrodes in both the common and uncommon conditions. Considering that theta-band oscillation and syllable processing are related, we propose that speakers created an abstract rhythmic pattern by combining the number of syllables with their appropriate temporal structure and the rhythmic priming effect observed in 400–800 ms time window originated from rhythmic processing before and during speech articulation in language production. Our results support the assumptions that speech prosody is not produced at a certain stage in speech production [15] and rhythmic encoding happens before articulation and during the articulation period in speech production.

### 5.1. Behavioral Evidence for the Rhythmic Priming Effect

Two previous studies investigated the rhythmic priming effect in language production at the behavioral level by using the reading-aloud task [8,22]. They both observed a rhythmic priming effect, with faster reaction times for targets that had the same rhythmic pattern as the preceding rhythmic prime. The present study again revisited the rhythmic priming issue while addressing the methodological limitations of the reading-aloud task. In the reading-aloud task, after the priming stimuli, the target speeches were visually presented and participants were instructed to speak the target speech aloud. The visually presented target speeches might have involved comprehension processing, which might make the production processing less “pure”. 

In this study, a phrase-recall paradigm was used to address the above problems, in which participants were required to recall the target phrases aloud after the priming stimuli. In line with our prediction, the experiment demonstrated a significant rhythmic priming effect, with reading latencies being shorter under conditions where the prime and target rhythmic patterns were congruent than under those where they were not, but only for the 2+2 rhythmic pattern. In 2+2 congruent situations, rhythmic priming facilitates speech production by reducing the time or resources required to prepare for the rhythmic patterns of the utterances. Our result is in line with previous speech production studies [8,13,14,21,22] and provides strong support for online rhythmic processing in speech production. In previous studies, by using the reading-aloud task, Gould et al. [22] found faster reaction times for target syllables that had the same rhythmic pattern as the preceding rhythmic prime. Similarly, Zhang and Zhang [8] also found that a certain rhythmic pattern could only influence the planning of speech output for the 2+2 rhythmic patterns, with reading latencies being faster in the congruent prime and target pattern situations. Here we need to note that in Zhang and Zhang’s study, the 2+2 target rhythmic pattern is a noun–verb combination, which is at the sentence level (e.g., 阿姨买菜, “the aunt buys vegetables”), while in our current study, the 2+2 target rhythmic pattern is a verb–noun combination, which is at the phrase level (e.g., 打扫房间, “clean the room”). Hence, we can conclude that in spite of different syntax materials and experimental tasks, the rhythmic effects in the former and current studies show very systematic patterns; participants can exploit the external rhythmic patterns and form abstract rhythmic information before articulation in language production. As for the underlying mechanisms of the rhythmic priming effect and the absence of the 1+3 rhythmic priming effect at the behavioral level, we will this discuss below in combination with the electrophysiological results. 

### 5.2. Cortical Tracking, Theta-Band Oscillation, and Rhythmic Priming Effect

The current study is the first to explore brain oscillations underlying the speech rhythm in language production. In line with our assumptions, we observed that the neural activity synchronizes with the rhythmic patterns of primes, and the cortical response to pure tones was comparable in terms of common and uncommon priming rhythmic patterns. The current results are consistent with dynamic attending theory [29,30,31], which assumes that the phase and the period of the internal rhythm’s oscillations are connected to an external rhythm when individuals attend to this rhythm. Though our task did not ask participants to attend to the rhythmic patterns in the prime stimuli, the strictly controlled rhythmic patterns (see Methods) could attract participants’ attention automatically. Similar results are also obtained in Harding et al. [33] and Nederlanden et al. [34] studies. Therefore, it is not surprising that there were no phase-locking differences between common (i.e., 2+2 prime rhythm) and uncommon (i.e., 1+3 prime rhythm) because the priming rhythm patterns were strictly controlled.

Based on the fact that the brain can track external rhythmic information, we hypothesized that a pure-tone priming rhythmic pattern could elicit temporal expectations about the speech rhythmic patterns of target speeches. Our results verified our hypothesis and showed that target phrases with congruent rhythmic patterns were associated with larger theta-band power (approximately 4–8 Hz) in the 400–800 ms time window in both the common (i.e., 2+2 rhythmic patterns) and uncommon (i.e., 1+3 rhythmic patterns) conditions. Participants may become aware of the priming rhythmic patterns after extensive repetition and tune into particular rhythmic patterns of the phrases over time. 

In the field of language studies, theta-band oscillation is often linked to the syllabic structure of speech mostly because the theta band oscillates at a similar rate to syllable production [5,47,50,51,70,71]. Furthermore, researchers have postulated a theta–gamma coupling mechanism, in which theta oscillations track the syllabic structure of speech and provide a temporal framework for grouping phonetic information transmitted by gamma oscillations [72,73,74]. In this study, we found increased theta-band power in target phrases that had congruent rhythmic patterns with the prime rhythm in the late window (400–800 ms) in both the common and uncommon conditions. This can be explained by the rhythmic pattern in Chinese. In Chinese speech production, syllables are proximate units of phonological encoding [75,76,77], and the rhythmic pattern is made up of words with varying syllable lengths [23]. Consequently, syllables can be seen as the basic unit for Chinese rhythm. When the priming rhythm is in accordance with the combination of words with different syllable lengths in the target stimuli (that is, in the congruent rhythmic condition), it may promote ease of production, which may reduce the time or resources needed to prepare for the rhythmic patterns of the utterances. 

Our results showed that the evoked theta-band power demonstrated that the congruent conditions of the rhythmic pattern facilitate the speech production process. A natural next step is to ask the following questions: (1) Why could the 1+3 rhythmic priming effect be detected in oscillational analysis while the 1+3 rhythmic priming effect was absent in the current behavioral analysis and our previous ERP studies? (2) What exact aspects of speech production are facilitated, and why?

For the findings of the 1+3 rhythmic priming effect in theta-band oscillations in the present study, a critical factor seems to be the repeated iterations of priming stimuli [8,21]. Researchers point out that the frequency of rhythmic patterns has an effect on speech perception and production [8,23]. As mentioned earlier in the introduction section, Chinese speakers generate more 2+2 rhythmic patterns (the common rhythmic pattern) than 1+3 rhythmic patterns (the common rhythmic pattern). Therefore, it was likely more challenging to establish neuronal synchronization for the 1+3 rhythmic pattern than for the 2+2 rhythmic pattern. In the current studies, the priming stimuli were repeated five times (in our former study, they were repeated three times), which might make it easier to form neural synchronization for the 1+3 rhythmic pattern. As a result, increasing the repeated times of the primes may be one candidate for the 1+3 priming rhythmic effect. 

However, an alternative explanation is that the 1+3 rhythmic priming effect may be related to the duration of pure tones. In our current study, each pure tone in the prime materials lasted for 250 ms so as to meet the average syllable duration in speech production. The matching time between the prime tones and real speech syllable duration may be another candidate for the 1+3 rhythmic priming effect. Further research can further investigate and compare these two possibilities. The absence of behavioral evidence for the 1+3 rhythmic priming effect is not unusual since electrophysiological methods are able to detect subtle brain processes that may not be detected by behavioral measures [1,78]. 

The next question concerns what aspects of speech production are facilitated. The average reaction time and the time window of the evoked theta-band power can provide some insight regarding this question. The average reaction time of the target speeches ranged from approximately 557 to 574 ms (see Figure 2) while the evoked theta-band neural oscillations occurred in the 400–800 ms time window. Clearly, the time window of theta-band neural oscillations outranged the average reaction time of the target speech. As the average reaction time of the target speeches mainly reflects the rhythmic processing before articulation, the theta-band neural oscillations that occurred in the time window of approximately 400–800 ms may correspond to the rhythmic encoding before articulation and during the articulation period in speech production. 

In our former research [8], we found that, before articulation, speakers have already established a prosodic abstract frame. By using the reading-aloud task, our former experiment found that in the 350–500 ms time window, the incongruent condition yielded a larger negative component (N400) than the congruent condition. Our current study provides evidence from neural oscillations that speakers need to encode rhythmic information before articulation. Based on the fact that neuronal oscillatory activity in the theta band (4–8 Hz) reflects syllable processing in speech [40], it is plausible to assume that for Mandarin Chinese (a syllable-timed language), the theta-band neuronal oscillations in our current study reflect the rhythmic encoding before articulation. 

So why does the theta-band neural oscillation also occur in the final stage of language production? According to prior models of working memory [79], we can divide the phrase-recall task in our experiment into three stages: Encoding, maintenance, and selection/retrieval. In the encoding stage (when the phrases were first given), the orthographic stimuli (four-word phrases) are mapped from a visual-orthographic to a phonological and auditory/motor representation. During the maintenance stage, the target utterance is rehearsed, and the activation of plans is stored in the memory. In the selection/retrieval stage (when participants were asked to speak the phrases), the plans are selected from memory and drive motor routines. As we mainly focus on the retrieval stage of the phrase-recall paradigm, it seems intuitively plausible that the rhythmic priming effect can also facilitate the execution of articulatory processing. Researchers also argue that syllables are meant to be the ideal articulatory motor units [12]. As we assume that syllables are the basic unit of speech rhythm, it is easier to understand why the priming effect occurs during the final stage of language production. 

Together with previous studies [8,13,14,21,22], our results support the assumption that speech prosody is not produced at a certain stage in speech production [15]. We propose a model of speech production in which an abstract level of prosody is incorporated (Figure 4). Prosodic information plays a crucial role in both the formulation and articulation processing stages. Signals from the conceptualization stage would convey information about a message that speakers intend to verbally express. Messages specify lexical concepts and their relationships. The message is then grammatically encoded, in which message properties are transferred onto one or more syntactic representations. Following that, speakers build sound-based representations that can encode segmental, metrical, or prosodic information through phonological encoding. Finally, speakers use articulatory mechanisms to move the muscles of the mouth and throat in order to present an expression to an audience. Prosodic information also influences articulation processing. Overall, such a model can explain the current studies’ findings. Given the scarcity of data in this area, additional research will be required to determine whether such a model can account for a broader range of findings.

## 6. Limitations and Future Directions

There are some limitations to this specific study that should be considered. First, it still remains an open question as to how best to quantify rhythmic processing in speech production. Though compared with the reading-aloud task, the phrase-recall priming paradigm adopted in the current study can reduce the involvement of the comprehensive process to a large degree, it could not detect the conceptualization stage in language production. Future research can adopt other experimental tasks such as the implicit priming paradigm [80], which can detect the full process of language production. Second, the participants who were recruited in the study all had no formal musical training background. Researchers argue that individual differences may affect speech rhythmic processing [33]. Future studies can recruit participants with musical backgrounds to determine whether musicians show similar patterns for speech rhythmic processing. The final limitation is the stimuli we used. Our current work used simple phrase-length utterances to explore rhythmic encoding in language production, and future studies can use a more challenging context such as poems or tongue twisters [58], where rhythmic priming effects may be more detectable.

## 7. Conclusions

To summarize, our research revealed a rhythmic priming effect on speech production based on behavioral and electrophysiological data. The theta band represented the anticipated rhythmic pattern in speech production. To the best of our knowledge, these findings provided the first brain oscillational evidence for the rhythmic priming effect in Mandarin phrase production. Furthermore, our results have shown that speakers form a prosodic abstract frame not only before articulation but also during the articulation stage in speech production. 

## Figures and Tables

**Figure 1 brainsci-12-01593-f001:**
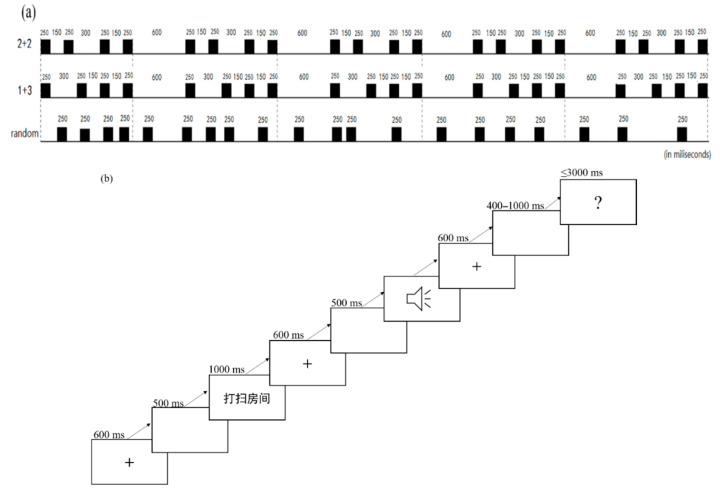
(**a**) Rhythmic patterns for primes (2+2 primes have a longer time interval between the second and third pure tone; 1+3 primes have a longer time interval between the first and second pure tone; random primes do not have a fixed time interval between tones). (**b**) A schematic of a trial.

**Figure 2 brainsci-12-01593-f002:**
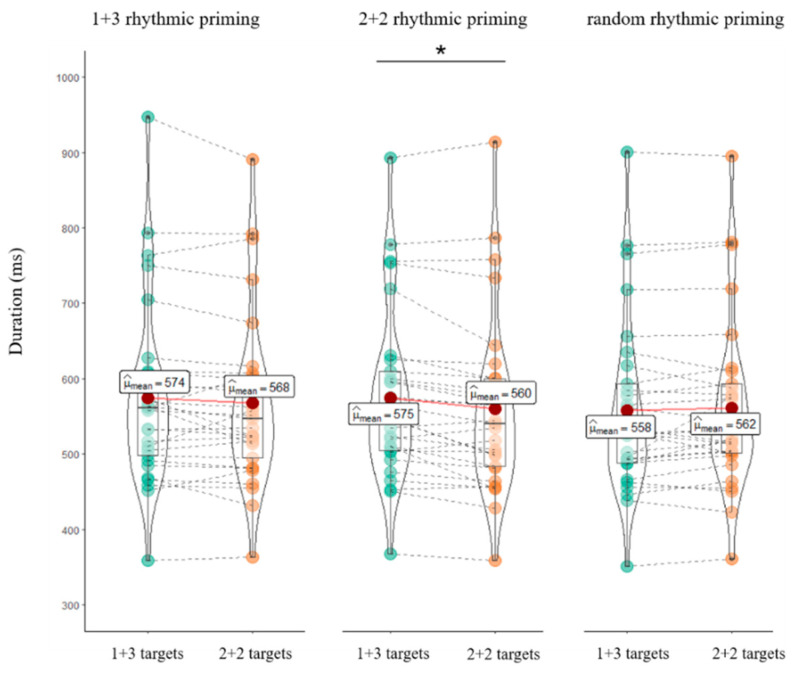
The differences in target conditions for each prime rhythm pattern across all of the participants. The points indicate averages for each participant; dark-red circles represent means; the solid red lines connecting the dark-red circles visually reveal the differences between the mean reaction time of different conditions; the dashed gray lines connecting the green or yellow points indicate scores for the same participant; and the boxplots represent the medians and interquartile ranges. * *p* < 0.05.

**Figure 3 brainsci-12-01593-f003:**
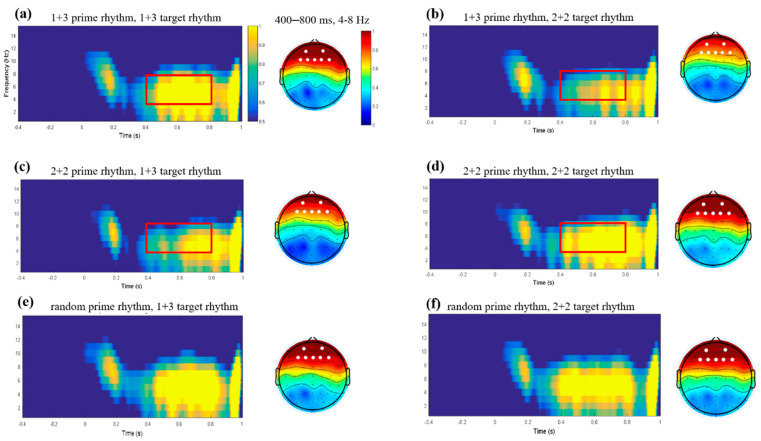
Time–frequency representations (TFRs) of power. (**a**,**b**) Grand average TFRs for the 2+2 target rhythm (**a**) and 1+3 target rhythm (**b**) under the condition of 1+3 prime rhythm; (**c**,**d**) grand average TFRs for the 2+2 target rhythm (**a**) and 1+3 target rhythm (**b**) under the condition of 2+2 prime rhythm; (**a**,**b**) grand average TFRs for the 2+2 target rhythm (**a**) and 1+3 target rhythm (**b**) under the condition of random prime rhythm. All theta power averaged over seven representative frontal channels (showed as white dots in topography). Topographical maps averaged in the time–frequency window of approximately 400–800 ms and 4–8 Hz, as indicated by the red–line–encircled region in each condition. (**g**) The differences in mean theta power averaged in this time–frequency window over frontal channels across all of the participants. The points indicate averages for each participant; dark–red circles represent means; the solid red lines connecting the dark–red circles visually reveal the differences between the mean theta power of different conditions; the dashed gray lines connecting the green or yellow points indicate scores for the same participant; and the boxplots represent the medians and interquartile ranges. * *p* < 0.05, ** *p* < 0.01.

**Figure 4 brainsci-12-01593-f004:**
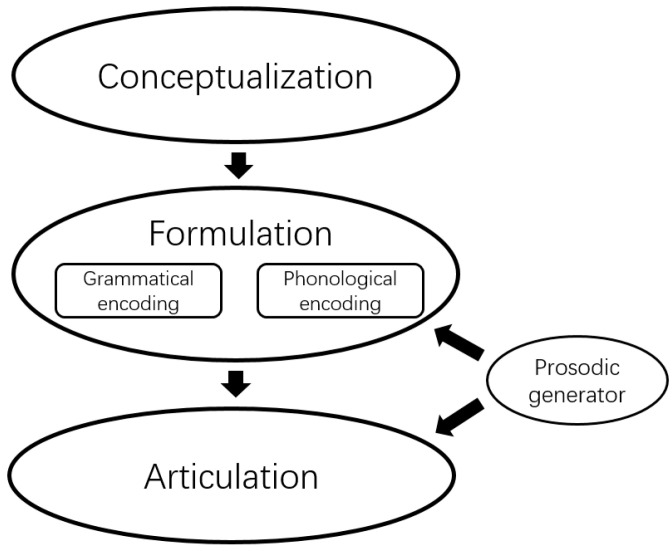
Model of speech production, including prosodic processing.

**Table 1 brainsci-12-01593-t001:** Fixed-effect ANOVA results for RTs.

Term	*Sum Sq*	*Mean Sq*	*df_Num_*	*df_Den_*
**Prime**	108,655.70	54,327.83	2	384.25
**Target**	32,927.37	39,279.37	1	375.31
**Prime × target**	66,479.60	33,239.80	2	384.25

Note. “Prime” indicates the rhythm type of prime (2+2, 1+3, random) and “Target” indicates the rhythm type of target speech (2+2, 1+3). *Sum Sq* indicates the sum of squares and *Mean Sq* indicates the mean of squares. *df_Num_* indicates degree of freedom numerator. *df_Den_* indicates degrees of freedom denominator.

## Data Availability

The data supporting the findings of this investigation are available upon reasonable request from the corresponding author. The data are not available to the public owing to anticipated additional analyses.

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
