# Peer review of "Theta Band (4–8 Hz) Oscillations Reflect Online Processing of Rhythm in Speech Production"

_brainsci, 2022, doi:10.3390/brainsci12121593_

Round 1
Reviewer 1 Report
This is indeed a very nice research about the processing of rhythm in speech production using Theta band of EEG signals. The paper is clear and the conclusions are supported by the reported results. I have found only a few issuses that should be addressed by the authors.
1. You have used EEG term in introduction (page 3, line 135) without introducing it.
2. I believe the authors can break the introduction into two sections. Introcution and liereature review. In this version, introduction is too long.
3. Page 4, line 192 (1.4. the present study). Please use capital T for the.
4. The authors are advised to make it more clear why AF3, AF4, F3, F4, F1, F2, Fz channels were selected for analysis.
Reviewer 2 Report
An introduction could have provided more information about the current study. This would have interested the reader in further reading.
The theoretical contribution of this article could be made more explicit. What gap is it intended to fill?
Is 11 students really a sufficient number for study participants? Please clarify this in the 'Materials and methods' section.
In the Discussion section, there is no real connection between this study and the theoretical basis and results of other similar studies.
The Conclusions do not state the significance of this study, the limitations, the directions for further research.
Round 2
Reviewer 2 Report
The article can now be accepted for publication.